Bone histology and growth curve of the earliest ceratopsian Yinlong downsi from the Upper Jurassic of Junggar Basin, Northwest China

Han Fenglu 1 hanfl@cug.edu.cn
Zhao Qi 2
Hu Jinfeng 1
http://orcid.org/0000-0002-4786-9948 Xu Xing 2 3
1 School of Earth Sciences, China University of Geosciences (Wuhan) , Wuhan , China
2 Key Laboratory of Vertebrate Evolution and Human Origins, Institute of Vertebrate Paleontology and Paleoanthropology, Chinese Academy of Sciences , Beijing , China
3 Centre for Vertebrate Evolutionary Biology, Yunnan University , Kunming , China
Moncunill-Solé Blanca
Electronic publication date: 2024 Dec 19
Publication date: 2024
Volume: 12
Electronic Location ID: e18761
Received 2024 Aug 14; Accepted 2024 Dec 4
Copyright: © 2024 Han et al.
Copyright year: 2024
Copyright holder: Han et al.
License: This is an open access article distributed under the terms of the Creative Commons Attribution License, which permits unrestricted use, distribution, reproduction and adaptation in any medium and for any purpose provided that it is properly attributed. For attribution, the original author(s), title, publication source (PeerJ) and either DOI or URL of the article must be cited.
License URL: https://creativecommons.org/licenses/by/4.0/

Keywords: Bone histology, Dinosaur, Ceratopsia, Jurassic, Junggar Basin, China, Yinlong, Growth curve

Funding: National Natural Science Foundation of China 41972021, 42288201, 42372046, 42072008, 42272020 This study was supported by the National Natural Science Foundation of China (41972021, 42288201, 42372046, 42072008, 42272020). The funders had no role in study design, data collection and analysis, decision to publish, or preparation of the manuscript.

==============================
Yinlong downsi, the earliest known ceratopsian, is represented by dozens of specimens of different sizes collected from the Upper Jurassic of the Junggar Basin, northwestern China. Here, we present the first comprehensive study on the bone histology of Yinlong downsi based on ten specimens varying in size. Four ontogenetic stages are recognized: early juvenile, late juvenile, subadult, and adult. The reconstructed growth curve suggests that Yinlong may reach sexual maturity at 6 years old, which is earlier than that of the well-studied early-diverging ceratopsian Psittacosaurus (9 years old) but later than ceratopsids (about 3 to 5 years old). This may indicate that sexual maturity begins earlier during the evolution of ceratopsians, and that the giant size of ceratopsids is acquired by accelerating growth rates. The cortex of the tibia mainly consists of fibrolamellar bone tissues, but parallel-fibered bone and lines of arrested growth (LAGs) are very common throughout ontogeny, suggesting a moderate growth rate. Quantitative analysis indicates that Yinlong has a maximum growth rate similar to those of other small-sized dinosaurs such as Psittacosaurus, Dysalotosaurus, and Troodon, and their maximum growth rates are higher than those of extant squamates and crocodiles but lower than those of extant mammals and large dinosaurs. This suggests that body size plays a more important role in growth rate than other factors such as phylogenetic position and/or diet among non-avian dinosaurs.

Introduction

Ceratopsia is a successful, large group of herbivorous non-avian dinosaurs that lived from the Late Jurassic to the end of the Cretaceous (You & Dodson, 2004). The early-diverging ceratopsians are all bipedal dinosaurs. They have rostral bones, relatively large skulls without horns, and relatively small body sizes, such as Yinlong, Psittacosaurus, Liaoceratops, Archaeoceratops, and Auroraceratops (Dong & Azuma, 1997; Xu et al., 2002; You & Dodson, 2004; You et al., 2005; Xu et al., 2006). Most of these materials are well studied in morphology and phylogeny (You & Dodson, 2003; Han et al., 2018; Morschhauser et al., 2018a, 2018c), but the ontogenetic variation of bone microstructure in early-diverging ceratopsians is still rarely known, except in Psittacosaurus. This latter species shows fibrolamellar bone with lines of arrested growth (LAGs) and was supposed to have faster growth rates than extant reptiles (Erickson & Tumanova, 2000; Erickson et al., 2009; Zhao et al., 2019; Skutschas et al., 2021). In addition, histological data support an ontogenetic shift from quadrupedality to bipedality (Zhao et al., 2013), and also suggest social behavior among different ages (Hedrick et al., 2014; Zhao et al., 2014).

Ontogenetic changes in bone microstructure of ceratopsians were also studied in Protoceratops, Cerasinops, Koreaceratops, the ceratopsids Einiosaurus, Pachyrhinosaurus, Centrosaurus, Avaceratops, Yuehecauhceratops, Utahceratops, Kosmoceratops and Triceratops (Chinnery & Horner, 2007; Reizner, 2010; Erickson & Druckenmiller, 2011; Horner & Lamm, 2011; Levitt, 2013; Fostowicz-Frelik & Słowiak, 2018; Hedrick et al., 2020; Baag & Lee, 2022; de Rooij et al., 2024). The Late Cretaceous Protoceratops is a small quadrupedal ceratopsian and possibly exhibits social behavior, as suggested for Psittacosaurus (Fastovsky et al., 2011). Protoceratops has a similar vascular pattern to Psittacosaurus but with no apparent LAGs and more fossilized fibers to strengthen the tissue (Fostowicz-Frelik & Słowiak, 2018). Bone microstructure of the non-coronosaurian neoceratopsians Koreaceratops is more similar to that of Protoceratops than to Psittacosaurus in preserving zonation of the tibia and LAGs in the fibula (Baag & Lee, 2022).

The large-sized ceratopsids are composed of fibrolamellar bone as in other ceratopsians, but the bone microstructure of centrosaurines and chasmosaurines is possibly different. The former is more like non-ceratopsid ceratopsians by preserving LAGs and lacks bone remodeling throughout the whole cortex, whereas the latter has few LAGs and strong bone remodeling throughout the cortex (Levitt, 2013; Hedrick et al., 2020; de Rooij et al., 2024). The centrosaurine Einiosaurus reaches its highest growth rate between 3 and 5 years, which is earlier than that of Psittacosaurus (Reizner, 2010). The humeri of Triceratops display a mix of longitudinal and circumferential vascular canals and bone remodeling happens in a very early stage, but LAGs appear very late (de Rooij et al., 2024).

Yinlong downsi is the earliest known ceratopsian recorded globally from the Upper Jurassic Shishugou Formation of Junggar Basin, China (Xu et al., 2006). It has a small body size (about 1–2 m in length) with a relatively large skull as in Psittacosaurus. Yinlong provides key features for understanding the evolution of early ornithischians, supporting the monophyly of Marginocephalia and its close relationship to Heterodontosauridae, although the latter is recovered as the most basal ornithischians in most phylogenetic analyses (Xu et al., 2006; Butler, Upchurch & Norman, 2008; Fonseca et al., 2024). Recent phylogenetic analyses support Yinlong to be within Chaoyangosauridae, which forms a monophyly with either Psittacosaurus or Neoceratopsia (Han et al., 2018; Morschhauser et al., 2018b; Yu et al., 2020). Abundant materials have been discovered allowing for a detailed study of skeletal morphology (Han et al., 2016, 2018), ontogeny, and dental system (Hu et al., 2022). However, there is still no detailed study on the growth pattern of Yinlong based on bone histology, hindering our understanding of the evolution of physiology and behavior in early ceratopsians and ornithischians. Here we provide an in-depth investigation to study ontogenetic variation and growth curvature and compare it with other ceratopsians and early diverging ornithischians, elaborating on the variation of growth patterns during the evolution of ceratopsians. In addition, bone microstructure from early ornithischians (Triassic and Jurassic) is relatively rarely known, and only a few taxa, such as Lesothosaurus (Knoll, Padian & de Ricqlès, 2010), heterodontosaurid remains (Becerra et al., 2016), have been primarily studied. The detailed ontogenetic study of bone histology in Yinlong will provide important evidence for understanding the evolution of physiology and behavior in early ornithischians.

Materials and Methods

Material

Ten individuals of Yinlong downsi were selected for histological analysis (Table 1). All the materials come from the Upper Jurassic Shishugou Formation of the Wucaiwan area, Xinjiang, China (Han et al., 2016, 2018). Identification of these materials is accurate if they have skulls, as Yinlong preserved many autapomorphies on its skull (Xu et al., 2006; Han et al., 2016). However, caution should be exercised if there are only postcranial bones, as they are quite similar to those of the small ornithopod Gongbusaurus wucaiwanensis (Dong, 1989) and another ceratopsian Hualianceratops wucaiwanensis (Han et al., 2015). Among these sampled ten individuals, eight have complete or partial skulls associated with postcranial elements, and only the two specimens IVPP V18677 and IVPP V18682 solely preserved the postcranial elements, but they have been well described in detail as Yinlong by Han et al. (2018). According to our observations, Yinlong differs from Gongbusaurus by its relatively robust femur, elongated ilium, blade-like prepubic process, and short postpubic shaft. Hualianceratops was another ceratopsian from the same stratigraphic layer as Yinlong (Han et al., 2015). Hualianceratops differs from Yinlong by its skull elements, but they have no differences in the postcranial materials. Hualianceratops was established only by one specimen, whereas Yinlong was represented by dozens of individuals. Here the preserved postcranial elements of IVPP V18677 and IVPP V18682 were possibly Yinlong, but Hualianceratops could not be excluded entirely.

Table 1 Basic dataset of sectional bones used for the calculation of growth curves.

Specimen number	Sectional bones	Femur circumferences (cm)	Femur length	Tibia length	Fibula length	Visible LAGs	Total LAGs	Body mass 1 (kg)	Body mass 2 (kg)	
IVPP V18677	Tibia	50	135	164	160	2 (1)	3	9	12	
Fibula	3 (0)	3	
IVPP V33266	Tibia	74	155 (E)	180	174	4 (1)	5	20	28	
Fibula	5	5	
IVPP V18678	Tibia	70	150 (E)	?	?	4 (1)	5	17	24	
IVPP V33267	Tibia	?	?	172	>147	5	5	?	?	
IVPP V18636	Tibia	86	178	?	?	4 (3)	7	31	42	
Fibula	5 (2)	7	
Rib	7 (0)	7	
IVPP V18679	Tibia	90	>170	>121	?	5 (2)	7	35	48	
Fibula	?	?	
Rib	4 (3)	7	
IVPP V18682	Fibula	?	177	196	>175	6 (2)	8	?	?	
Rib	6 (2)	8	
IVPP V18683	Tibia	97	189	210	198	8 (4)	12	?	?	
Fibula	9 (2)	11	
Rib	10 (1)	11	
IVPP V33268	Fibula	100	>164	214	>167	?	?	46	64	
IVPP V18637	Tibia	100	?	201		11 (2)	13	46	64	
Note:

“E” denotes estimated values. The numbers in parentheses denote the estimated erosion LAGs. Body mass 1 and 2 correspond to the studies by Anderson, Hall-Martin & Russell (1985) and Campione et al. (2014), respectively.

All the materials were photographed and measured before sectioning. The preparation of most of these sections (IVPP V18677, IVPP V18678, IVPP V18636, IVPP V18679, IVPP V18682, IVPP V18683, IVPP V18637) was performed in the IVPP (Institute of Vertebrate Paleontology and Paleoanthropology). These samples were processed using the Exakt-Cutting-Grinding System (Donath & Rohrer, 2003), and the detailed method was provided in Han et al. (2020). The other sections (IVPP V33266, IVPP V33267, IVPP V33268) were prepared at the fossil lab, School of Earth Sciences, China University of Geosciences (Wuhan) (Ke et al., 2021). Sections were sampled mainly from the mid-diaphyses of the tibia, fibula, and ribs (Table 1; Fig. S1). The sectional sides were photographed using a Zeiss Primotech microscope. Throughout the text, bone histological terminology and definitions follow Francillon-Viellot et al. (1990) and Chinsamy-Turan (2005).

Body mass estimation

The body mass of bipedal dinosaurs is usually estimated by the following two equations:

Massbiped=0.16FC2.73 (Anderson, Hall-Martin & Russell, 1985).

Massbiped=(10(2.749∗log10(FC∗20.5)−1.104))/1,000 (Campione et al., 2014; Benson et al., 2018).

FC indicates the minimum circumference of a femur, which has been acquired in seven individuals (Table 1).

Longevity and growth rate calculation

Estimation of individual age was calculated by counting LAGs in tibiae, fibulae, and ribs (Fig. 1). We preferred to count the LAGs in the ribs for relatively small individuals (IVPP V33266, IVPP V18636), which usually preserved the most complete LAGs because of a small medullary cavity and less remodeled than the fibulae and tibiae (Waskow & Sander, 2014; Fostowicz-Frelik & Słowiak, 2018; Hedrick et al., 2020; de Rooij et al., 2024). Therefore, longevity could be directly estimated by preserving LAGs in small individuals. However, in large individuals, inner bone erosion happened in all sectional bones, and longevity was estimated referring to the estimated LAGs of the tibia, fibula, and ribs. Missing LAGs destroyed by the expansion of the medullary cavity are recalculated by using the extrapolation method (Klein & Sander, 2007). The distance between the center of medullary cavity and the first visible LAG was divided by the greatest distance between any two adjacent LAGs. The result of the calculation is the maximal number of estimated LAGs as the resorbed cycle is wider than any preserved ones (Table 1; Fig. S2). Therefore, it is more accurate to subtract one. The total estimated LAG was calculated using the following equation:

Nest=Npre+(Dcf/Dmax)−1

Figure 1 Skeleton and bone microstructure of the subadult individual IVPP V18636.

(A) Skull and postcranial skeleton in dorsal view. The black arrow denotes the sampled position. Cross-section (B) and line drawing LAGs (C) of the right tibia. Abbreviations: dv, dorsal vertebrae; mc, medullary cavity; rfe, right femur; rfi, right fibula; ril, right ilium; rsc, right scapula; sk, skull.

Nest denotes the total estimated LAGs. Npre denotes preserved LAGs. Dcf means the distance between the center of the medullary cavity to the first LAGs. Dmax means the greatest distance between the preserved adjacent LAGs.

Growth rate estimation of consecutive years was obtained by subtracting body mass values. To compare with the maximal growth rates of extant animals, the rates were then converted to daily values by dividing each by 379 days in 1 year of the Late Jurassic (Wells, 1963). The converted daily data value was compared to the data of extant vertebrate taxa of similar size (Erickson, Rogers & Yerby, 2001; Hübner, 2012).

Growth curves

Here we established the growth curves according to Erickson & Tumanova (2000). The estimated body mass was plotted against their age in years. Establishing the growth curves was analyzed using the statistical software Origin 8.0. The equation mass = a/(1+ exp (b(Age + c))) was chosen to describe the sigmoidal growth curve (a, largest known body mass; b, c denote parameters to fit). As mentioned above, we used two methods to estimate the body mass and got two growth curves. The body mass estimation by Anderson, Hall-Martin & Russell (1985) was mainly based on extant mammals ratites, whereas Campione et al. (2014) analyzed more birds and got significantly lower errors. However, the formula of Anderson, Hall-Martin & Russell (1985) is commonly used for estimating the body mass of non-avian dinosaurs before the publication of Campione et al. (2014) and there is more data on the growth curve based on the former. Here we used both of these methods for body mass estimation to establish growth curves.

To compare the maximum growth rate of Yinlong with other dinosaurs and extant animals, we used the dataset of Grady et al. (2014). The regression of adult body mass and maximum growth rates of crocodilians, placental mammals, marsupials, altricial birds, precocial birds, squamates, and dinosaurs were calculated and used for comparison. The body mass estimation of Yinlong was based on Anderson, Hall-Martin & Russell (1985) for consistency with other dinosaurs.

Results

The bone histology of Yinlong downsi consists of fibrolamellar bone tissue with high cortical porosity, but the parallel fibered bone tissue and LAGs are common throughout ontogeny, suggesting that Yinlong has a moderate growth rate. Here, we divided the growth of Yinlong into four stages, including early juvenile, late juvenile, subadult, and adult, based on a combination of features of bone microstructure. The description of each stage is mainly based on the tibia. Fibulae and ribs are also described for comparison.

Early juvenile stage

IVPP V18677 is the smallest individual, with a femur length of approximately 135 mm (Table 1). The bone microstructure of the tibia and fibula was examined and corresponds to the early juvenile stage (Fig. 2). The sampled partial cortex of the tibia is highly vascularized, with most vascular canals oriented longitudinally and only a few anastomosing canals present (Fig. 2C). Primary osteons are well developed throughout the cortex, while secondary osteons are absent at this stage. Two indistinct LAGs are present, dividing the cortex into three circumferential regions (Fig. 2A). Under polarized light, the cortex mainly consists of fibrolamellar bone, but parallel-fibered bone tissue is prevalent throughout the cortex, and increases in density towards the LAG and the bone surface (Figs. 2B, 2D), suggesting alternating rapid and slow growth rates. Notably, both the inner and outer regions of the cortex exhibit dark colors under normal and polarized light. At higher magnification, dense oval-shaped osteocytes with complex canaliculi are clearly visible in the outer region of the tibial cortex (Fig. 2E). Osteocyte lacunae are abundant and randomly distributed throughout the cortex, yet they are organized in lines parallel to the LAGs. In the fibula, vascular density is relatively low compared to that of the tibia (Fig. 2F). The partial cortex consists of parallel-fibered bone tissue with longitudinal vascular canals (Fig. 2G). No bone remodeling is observed in the inner region. Three LAGs are present, indicating that the innermost LAG of the tibia may have been eroded due to the enlargement of the medullary cavity. The medullary cavity of the fibula is small.

Figure 2 Bone microstructure of the tibia and fibula in early juvenile Yinlong downsi IVPP V18677.

(A–E) Tibia. Bone microstructure of tibia under normal (A) and polarized (B) light. (C) Enlargement of the mid-cortex showing vascularization type and LAGs. (D) Enlargement of the outer cortex showing parallel-fibered bone tissue and the outer LAG. (E) Enlargement of the outer cortex showing osteocyte and dense canaliculi. (F, G) Fibula. Parallel-fibered bone tissue with longitudinal vascular canals under normal (F) and polarized (G) light. All arrows indicate LAGs. Abbreviations: vc, vascular canal; ol, osteocyte lacuna.

Late juvenile stage

The late juvenile stage is represented by the tibia of IVPP V33266 (Table 2, Fig. 3), which has a femur length of approximately 155 mm. The vascular density throughout the cortex is notably high. The vascular organization is primarily characterized by reticular vascularization, although longitudinal vascularization is also abundant (Fig. 3A). Most vascular canals are encased in a thin layer of lamellar bone, forming primary osteons, while a few simple primary vascular canals and open vascular canals are observed near the periphery (Figs. 3B, 3C). No secondary osteons are present. Fibrolamellar bone tissue is predominant, but parallel-fibered bone is also common and highly organized near the LAGs (Figs. 3A, 3B). Four LAGs are visible circumferentially in the cortex, with the outermost LAG accompanied by a thin layer of highly organized parallel-fibered bones near the periphery (Figs. 3A, 3B). Osteocyte lacunae are oval or nearly round throughout most of the cortex, but they appear relatively flattened near the LAGs and in the outermost region. Sharpey’s fibers are present in the outer cortex of the tibia (Fig. 3A). Bone tissue of the fibula consists of fibrolamellar bone with predominant longitudinal vascular canals, but a few secondary osteons are observed in the innermost region (Figs. 3E, 3F). The sectional dorsal rib exhibits less bone remodeling compared to limb bones. It contains a small medullary cavity, a thin layer of endosteal bone, and five complete LAGs, indicating that the individual was at least 5 years old when it died (Fig. 3D).

Table 2 Characteristic bone microstructure of the tibia in the three growth stages of Yinlong downsi.

	Early juvenile	Late juvenile	Subadult	Adult	
Specimen number	IVPP V18677	IVPP V33266; IVPP V18678	IVPP V18636; IVPP V33267; IVPP V18679; IVPP V18682	IVPP V18683; IVPP V33268; IVPP V18637	
Primary osteons	Abundant	Abundant	Abundant	Abundant	
Erosion cavity in the inner cortex	No	Yes	Yes	Yes	
Secondary osteons	No	No	Few to abundant	Dense	
Secondary remodeling	No	No	Yes	Yes	
Endosteal bone	Absent	Absent or thin	Absent or thin	Absent or thin	
Dominated bone tissue type	Fibrolamellar bone	Fibrolamellar bone	Fibrolamellar bone and Parallel-fibered bone	Fibrolamellar bone and parallel-fibered bone	
Thickened and highly organized parallel-fibered bone in the outer cortex	No	No	Yes	Yes	
Vascular canal shape	Longitudinal	Longitudinal and reticular	Longitudinal and reticular	Reticular	
Preserved LAG number	2	4–5	4–7	8–10	
Total LAGs	1–3	4–5	6–10	>11	
Note:

Total LAGs denote the preserved LAG plus the estimated erosion LAGs.

Figure 3 Bone microstructure of the limb bones in late Juvenile Yinlong downsi (IVPP V33266).

(A–C) Tibia. Partial cortex under normal (A) and polarized (B) light; (C) Enlargement of the outermost cortex showing parallel-fibered bone and one vascular canal. (D) Partial cortex of one dorsal rib, showing LAGs. (E, F), Fibula. Bone microstructure in the innermost region under normal (E) and polarized (F) light, showing dense secondary osteons and partial endosteal bone tissue. All arrows indicate LAGs. Abbreviations: ira, irregular vascular canal; lvc, longitudinal vascular canal; sf, Sharpey’s fibers; so, secondary osteons.

IVPP V18678 is similar in size to IVPP V33266 (Table 1). The whole cross sections of the tibia and fibula were sampled. The cortex of the tibia is elliptical, with a maximum diameter of 16.5 mm and a minimum diameter of 13.5 mm (denoted as 16.5 × 13.5 mm). It consists of fibrolamellar bone with predominant longitudinal primary osteons (Figs. 4A, 4C). Four LAGs are visible in the cortex under normal light. Alternating woven bone (dark color) and parallel-fibered bone (bright color) are distinctly observable under polarized light (Fig. 4B). The distance between adjacent LAGs is relatively equal. Differing from IVPP V33266, the inner region of the tibial cortex forms many large resorption cavities and a thin layer of endosteal bone (Figs. 4C, 4D). In the fibula, dense secondary osteons are observed throughout the whole cortex (Figs. 4E–4G), and a thick layer of endosteal bone (about 100 μm) has formed in the innermost region (Figs. 4F, 4G).

Figure 4 Bone microstructure of the limb bones in late Juvenile Yinlong downsi IVPP V18678.

(A–D) Tibia. Partial cortex under normal (A) and polarized (B) light. (C) Enlargement of the mid cortex under normal light, showing longitudinal vascular canals. (D) Enlargement of the inner cortex under polarized light, showing erosion cavities. (E–G) Fibula. (E) Partial cortex under normal light. Enlargement of the inner cortex under normal (F) and polarized (G) light, showing dense secondary osteons and a wide endosteal layer. Abbreviations: eb, endosteal bone; ec, erosion cavity; pf, parallel fibered bone; wf, woven-fibered bone. All arrows indicate LAGs.

Subadult stage

The subadult stage is represented by three specimens. IVPP V18636 has a femur length of about 178 mm (Table 1). The tibia, fibula, and rib were sectioned. The cortex of the tibia is elliptical, with a maximum diameter of 22.4 mm and a minimum diameter of 14.7 mm (Fig. 5). Most vascular canals are longitudinal, and there are also many irregular vascular canals (Figs. 5A–5D). The cortex consists of a large area of parallel-fibered bone tissue in its outer region (Fig. 5D). Four LAGs are distributed in the cortex, with the innermost two LAGs closely positioned, forming double LAGs (Fig. 5A). The distance between adjacent LAGs decreases towards the periphery. Remodeling of secondary osteons occurs in the innermost cortex (Figs. 5E, 5F).

Figure 5 Bone microstructure of the limb bones in subadult Yinlong downsi IVPP V18636.

(A–F) Tibia. Partial cortex under normal (A) and polarized (B) light. Enlargement of the outer cortex showing LAGs and parallel-fibered bone under normal (C) and polarized (D) light. Bone microstructure of the innermost region under normal (E) and polarized (F) light, showing dense secondary osteons. (G) Partial cortex of the fibula. (H) Whole cross-section of the dorsal rib. All arrows indicate LAGs. Abbreviation: so, secondary osteon.

The whole cross-section of the fibula is elliptical. Most of the vascular canals are longitudinal and irregular (Fig. 5G). The cortex consists of fibrolamellar bone tissue in the middle and lateral regions, but remodeling of dense secondary osteons is present in the inner region (Fig. 5G). The cortex of the rib mainly consists of a parallel fibered bone matrix. Seven LAGs are visible (Fig. 5H). Secondary osteons are distributed in the inner region, medial to the second LAG. The medullary cavity of the rib is very small and it seems that there is no erosion of LAG, suggesting that the individual was 7 years old when it died.

In IVPP V18679, the whole cross-section of the tibia is subtriangular (20.6 × 17.4 mm). The average thickness of the cortex is about 3.7 mm. The medullary cavity is very large. The vascular canals are abundant and irregular, forming reticular fibrolamellar bone tissue (Fig. 6A). Primary osteons are abundant, and dense secondary osteons concentrate at the corner of the cross section (Figs. 6C, 6D). Additionally, vascular canals near these secondary osteons are parallel and extend toward the corner (Fig. 6C). Five LAGs are distinctly visible, with their distances gradually decreasing toward the periphery (Figs. 6A, 6B). The outermost two LAGs are very close to each other. The cross section of a dorsal rib from IVPP V18679 is subcircular (6.7 × 4.3 mm). Vascular canals are abundant and longitudinal. The inner region contains several large erosion cavities (Fig. 6D), and a few secondary osteons are present. Four LAGs are clearly observable. A thin layer of endosteal bone is evident in the innermost region.

Figure 6 Bone microstructure of the tibia and rib in subadult Yinlong downsi IVPP V18679.

(A–D) Tibia. Partial cortex under normal (A) and polarized (B) light. Bone remodeling in the inner cortex under normal (C) and polarized (D) light. (E) Whole cross-section of the dorsal rib. All arrows indicate LAGs. Abbreviations: ec, erosion cavity; so, secondary osteon.

In IVPP V18682, the whole cross section of the fibula is elliptical (9.3 × 5.5 mm) (Fig. 7A). Vascular canals are mainly longitudinal. Primary osteons are distributed in the mid and outer regions. Dense radial canals are present in the outer cortex, surrounded by a thin layer of lamellar bone tissue (Fig. 7). The dense radial canals are similar to the pathological bone microstructure of an extant turkey vulture and non-avian dinosaur from Transylvania (Chinsamy & Tumarkin-Deratzian, 2009). A few secondary osteons and large erosion cavities are observed in the inner region (Figs. 7D, 7E). Seven LAGs are shown in the cortex. The inner six LAGs are evenly spaced and the seventh LAG is distributed near the periphery.

Figure 7 Bone microstructure of the fibula in subadult Yinlong downsi IVPP V18682.

(A) Whole cross-section; Enlargement of the dense radial canals (likely pathological bone) under normal (B) and polarized (C) light. Enlargement of the innermost region under normal (D) and polarized (E) light. All arrows indicate LAGs. Abbreviations: eb, endosteal bone; ec, erosion cavity.

Adult

The adult stage was represented by two individuals, IVPP V18683 and IVPP V18637 (Table 2). Vascular canals are reticular at the middle region but predominantly longitudinal in the outer region (Fig. 8A). Vascular canals decrease in number towards the periphery, and scarce vascular canals are shown near the periphery, suggesting a very slow growth rate at the time of death (Figs. 8E, 8F). The cortex of the tibia mainly consists of fibrolamellar bone in the mid-region and a thick layer of parallel-fibered bone in the outer region (Figs. 8A, 8B). Eight and ten LAGs are visible in the cortex of IVPP V18683 and IVPP V18637, respectively. The distance between adjacent LAGs is very close near the periphery (Fig. 8A). Dense secondary osteons are observed in the inner cortex (Figs. 8C, 8D). A thin layer of endosteal bone is observed in the innermost cortex of IVPP V18683 but is absent in that of IVPP V18637. The medullary cavity is relatively large.

Figure 8 Bone microstructure of limb bones and rib in adult Yinlong downsi.

(A–D) Tibia of IVPP V18683. Partial cortex under normal (A) and polarized light (B). Enlargement of the innermost region showing dense secondary osteons under normal (C) and polarized light (D). Tibia of IVPP V18637 under normal (E) and polarized (F) light. (G, H) Fibula of IVPP V18683. (G) Partial cross section under normal light, showing longitudinal vascular canals and LAGs. (H) Outermost region under polarized light showing lamellar bone tissue without vascular canal. (I–K) Rib of IVPPV18683. (I) Whole cross secion of a dorsal rib. Partial bone section showing LAGs and secondary osteons in outer cortex under normal (J) and polarized (K) light. All arrows indicate LAGs. Abbreviation: so, secondary osteon.

The fibula of IVPP V18683 preserved nine LAGs (Fig. 8G). Vascular canals are predominantly longitudinal. The periphery consists of three closely packed LAGs, forming a wide lamellar bone devoid of vascular canals (Figs. 8G, 8H). The cortex of the rib mainly consists of parallel-fibered bone tissue (Figs. 8I–8K). The vascular canals are longitudinal and decrease in number towards the periphery. The inner region consists of dense secondary osteons, while only a few secondary osteons are observed in the outer cortex (Figs. 8J, 8K). Ten LAGs are visible in the cortex and are tightly articulated (Fig. 8J). The inner LAGs are partially erased by secondary osteons. The distance between adjacent LAGs decreases toward the periphery, with the outermost four LAGs being in very close proximity to each other.

Discussion

Variation of bone tissue in Yinlong downsi

Generally, all of the sampled sections display parallel-fibered “fibrolamellar” bone that is interrupted by LAGs, as in many small dinosaurs, such as Jeholosaurus, Lesothosaurus Orodromeus, and Psittacosaurus (Horner et al., 2009; Knoll, Padian & de Ricqlès, 2010; Zhao et al., 2019; Han et al., 2020). The parallel-fibered “fibrolamellar” bone is also reported in Triceratops, in which the bone matrix is not strictly woven but is often characterized by parallel-fibered bone (de Rooij et al., 2024). Similar bone tissue defined as “modified laminar bone” is observed in some sauropods, indicating relatively slower growth rates than that of typical fibrolamellar bone (Klein et al., 2012).

The variation of bone tissue is observed during ontogeny and different skeletal elements. For the tibia, the juvenile stage generally has a longitudinal vascular pattern, followed by the subadult stage with a reticular vascular pattern and dense secondary osteons in the inner part of the cortex. The adult is marked by thick parallel fibered bone with fewer vascular canals and closely compact LAGs but no EFS (external fundamental system). Similar ontogenetic variations are also observed in most small to medium-sized dinosaurs (e.g., Jeholosaurus, Psittacosaurus), but vary in different taxa. In Psittacosaurus, secondary remodeling happened in the relatively late growth stage, and endosteal bone is more complete in the adult (Zhao et al., 2019). Specifically, parallel-fibered bone tissue is prominent through ontogeny and an EFS is present in the small theropod dinosaur Masiakasaurus (Lee & O’Connor, 2013).

For the fibula and ribs, the vascular pattern is mainly longitudinal. Bone remodeling happens earlier in the fibula than in the tibia. This differs from the condition observed in Jeholosaurus, in which the fibula is less remodeled and preserves the complete endosteal bone and LAGs (Han et al., 2020).

Both the tibia and fibula of Yinlong have relatively larger medullary cavities than those of the ribs. The dorsal ribs are less remodeled and preserve the most complete number of LAGs compared to other elements, making them better for age estimation than other elements. This is also the case that was proved in many dinosaurs and crocodylomorphs, and the proximal end of the rib shaft is the best sample position and preserved the nearly complete growth record (Waskow & Sander, 2014; Waskow & Mateus, 2017). However, age estimation based on dorsal ribs may not be very accurate in adult individuals. For example, in the adult individual of Yinlong, the cortex of the dorsal rib is composed of lamellar bone, and many LAGs and other growth marks are concentrated near the periphery and sometimes they are not easy to discern (Fig. 8). In addition, strong bone remodeling obscuring LAGs was found in the largest rib of Pachyrhinosaurus (Hedrick et al., 2020). Therefore, it is not enough to estimate the longevity in the late ontogenetic stage only according to the bone microstructure of the dorsal rib. Cross sections of limb bones such as the femur, tibia, and fibula, are also necessary for accurate age estimation.

Endosteal bone is more common in the fibulae than other elements. In most elements, the distance of adjacent LAGs gradually decreases towards the periphery. However, double LAGs are present in the tibia of subadult IVPP V18636. Double or triple LAGs are commonly present in extant and extinct taxa, including amphibians, reptiles, birds, and mammals (Cullen et al., 2021; Han, Zhao & Liu, 2021). These growth marks are usually considered to be formed in 1 year. The occurrence of multiple LAGs was regarded as a harsh environmental condition or aestivation (Köhler & Moyà-Solà, 2009; Sanchez et al., 2010; Cullen et al., 2021). However, sometimes the double LAGs are not easy to identify, especially near the periphery. Cullen et al. (2021) suggested tracing the full circumference of a section to confirm if they were completely distinct, but a whole cross-section is difficult to make in large bones such as sauropods, hadrosaurs, and ceratopsids. The longevity of a taxon could be overestimated if the multiple LAGs are not recognized correctly. A thick layer of porous radial canals are seen in a fibula of Yinlong (Fig. 7), and they are also seen in the bone tissue of tibiae of P. mongoliensis at approximately 8 years of age (Erickson & Tumanova, 2000). However, dense radial canals are not seen in any normal bone tissue of P. lujiatunensis and Koreaceratops but are observed in the fractured bone of Psittacosaurus, and were regarded as a pathology (Chinsamy & Tumarkin-Deratzian, 2009; Hedrick et al., 2016; Zhao et al., 2019). Nevertheless, dense radial canals were also reported in the tibia of a late growth stage in P. mongoliensis and were supposed to reflect osseous drifting (Erickson & Tumanova, 2000).

The life history of Yinlong

The overall growth pattern of Yinlong is S-shaped as in most extant vertebrates and other dinosaurs (Fig. 9) (Zullinger et al., 1984; Erickson, Rogers & Yerby, 2001). The perinatal ontogenetic stage is not preserved in Yinlong. According to body mass estimation by Campione et al. (2014), the growth rates for Yinlong are up to 17.4 g/day at the age of three, increased up to 24.6 g/day at 5 years old, and gradually decreased to 1.6 g/d at the age of 12. The maximum growth rate is 14.6 g/day based on body mass estimation by the formula of Anderson, Hall-Martin & Russell (1985). Yinlong reaches its maximum growth rates at the juvenile stage as in other dinosaurs. The maximum growth rate of Yinlong is similar to that of other small dinosaurs, such as Psittacosaurus (13.8 g/day) (Erickson & Tumanova, 2000), Dysalotosaurus (18.6 g/day), Coelophysis (11.2 g/day), Saurornitholestes (14.1 g/day), and Troodon (16.6 g/day) and is faster than most extant reptiles but slower than large-sized dinosaurs, extant mammals and avian taxa (Fig. 10) (Hübner, 2012; Grady et al., 2014). The body size of ceratopsians increased during their evolution but the number of LAGs did not increase correspondingly but decreased in late diverging ceratopsids, suggesting that the maximum growth rate was accelerating during the evolution of ceratopsians.

Figure 9 Growth curves (in red) of Yinlong downsi.

The sigmoidal equation is used here to describe its growth pattern. (A) Body mass estimated by Campione et al. (2014); (B) Body mass estimated by Anderson, Hall-Martin & Russell (1985). Abbreviations: SEX, sexual maturity; SOM, somatic maturity.

Figure 10 Comparison of the maximum growth rate of Yinlong with other dinosaurs and extant animals.

Yinlong is located at the regression line of dinosaurs.

In Yinlong, sexual maturity occurs at approximately 6 years old based on a slowdown of growth rate (Fig. 9) and correlates with a transition of bone tissue from fibrolamellar bone to a parallel-fibered bone in the outer cortex seen in subadult and adult stages (Figs. 5–8). This is earlier than those of Psittacosaurus (Erickson et al., 2009) and some iguanodontians, such as Dysalotosaurus (Hübner, 2012) and Tenontosaurus (Lee & Werning, 2008), but older than the ceratopsid Einiosaurus (3–5 years old) (Reizner, 2010), many small ornithopods such as Jeholosaurus (Han et al., 2020), Orodromeus (Horner & Goodwin, 2009) and hadrosaur Maiasaura (Woodward et al., 2015). It seems that the attainment of sexual maturity is earlier in ceratopsids than in early-diverging ceratopsians. However, in ornithopods, the attainment of sexual maturity is delayed from small ornithopods to iguanodontians but is earlier in hadrosaurs (Lee & Werning, 2008; Woodward et al., 2015; Han et al., 2020). In summary, both these clades have earlier sexual maturity in late-diverging taxa.

The presence of EFS usually indicates a fully grown stage in vertebrates and has been found in extant mammals, pterosaurs, birds, and all kinds of non-avian dinosaurs, such as Troodon, Europasaurus, Apatosaurus, Maiasaura, and Stegosaurus (Varricchio, 1993; Horner, De Ricqles & Padian, 2000; Chinsamy-Turan, 2005; Sander et al., 2006; Klein & Sander, 2008; Hayashi, Carpenter & Suzuki, 2009; Bertozzo et al., 2021). EFS was also found in Pseudosuchia lineage, such as the American alligator and the aetosaur Aetosauroides scagliai, suggesting that determinate growth is shared by Archosauria (Woodward, Horner & Farlow, 2011; Rainwater et al., 2022; Ponce, Desojo & Cerda, 2023). The absence of EFS in our thin sections suggests that the largest sampled individual of Yinlong is still not fully grown. However, the closely-spaced LAGs and parallel-fibered bone all indicate nearly maximum body size.

Comparison of bone microstructure between Yinlong and other ceratopsians

Psittacosaurus is more derived than Yinlong, and the bone histology of three species of the former has been well studied, including P. mongoliensis, P. lujiatunensis and P. sibiricus (Erickson & Tumanova, 2000; Zhao et al., 2019; Skutschas et al., 2021). They all consist of fibrolamellar bone with predominant longitudinal and reticular vascular canals in tibiae and fibulae, as in Yinlong (File S2). Thick parallel fibered bone layers are shown in the outer cortex of the late ontogenetic stage (subadult) of P. lujiatunensis and P. sibiricus, but it is absent in the largest individual of P. mongoliensis (estimated to be about 9 years old) (Erickson & Tumanova, 2000). In addition, erosion bays are present in the early stage of the tibiae in Yinlong, P. lujiatunensis and P. sibiricus, but are only observed in the tibia of an eight-year-old specimen in P. mongoliensis (Erickson & Tumanova, 2000). Therefore, the bone microstructure of Yinlong is more similar to those of P. lujiatunensis and P. sibiricus than P. mongoliensis, and the largest known individual of P. mongoliensis is still histologically immature. However, secondary remodeling is shown in the subadult stage of Yinlong, as in P. sibiricus, but it only appeared in the largest individual of P. lujiatunensis (Zhao et al., 2013, 2019; Skutschas et al., 2021).

Radiating reticular vascularization, which is suggested to have rapid growth rates, was observed in the limb bones of Psittacosaurus, but is absent in tibial bone sections of Yinlong, suggesting the latter has a relatively lower growth rate. A thin layer of endosteal bone is present in the tibia of Yinlong, and it is thicker in the fibula, as in the condition of P. lujiatunensis (Zhao et al., 2019). However, no endosteal bone is found in the largest specimen of Yinlong, unlike that of P. lujiatunensis, which preserved a complete endosteal bone in adult individuals (Zhao et al., 2019). It seems that the endosteal bone may be eroded by the expansion of the medullary cavity in the tibia of adult individuals of Yinlong. LAGs are clearly observed in the early stage of Yinlong, as in Psittacosaurus (Erickson & Tumanova, 2000; Zhao et al., 2013, 2019) and many other small dinosaur bones (Chinsamy-Turan, 2005; Han et al., 2020).

Likewise, Koreaceratops is a small non-coronosaur neoceratopsian and more derived than Yinlong and Psittacosaurus (Lee, Ryan & Kobayashi, 2011; Morschhauser et al., 2018b). The length of preserved caudal vertebrae and partial pelvic girdle is less than 1 m, suggesting a small body size, but it may reach the subadult stage according to bone microstructure (Lee, Ryan & Kobayashi, 2011; Baag & Lee, 2022). The fibula shows similar longitudinal and reticular vascular canals as those of Yinlong and Psittacosaurus. However, the tibia of Koreaceratops consists of longitudinal vasculature with circumferentially elongated osteon canals, and no LAG is observed. Instead, alternative zones and annuli is detected in the tibia. LAGs are present in the outermost part of the Koreaceratops right fibula. The degree of bone remodeling in the Yinlong tibia is lower than in the fibulae, as in the condition of Koreaceratops (Baag & Lee, 2022). Massive remodeling is reported in the Koreaceratops fibula which may be a result of a thin section near the proximal end. Osseous drift happens in the fibula of Yinlong, and it is also present in Psittacosaurus (Erickson & Tumanova, 2000; Zhao et al., 2019) and Koreaceratops (Baag & Lee, 2022), but the latter have heavier bone drift by the presence of anteromedial erosion rooms deep into the mid-cortex.

Cerasinops is a small non-coronosaur neoceratopsian that is closely related to Koreaceratops. It is possibly bipedal as in other early diverging ceratopsians (Chinnery & Horner, 2007). Both the hindlimb and forelimb of Cerasinops were sampled to study bone microstructure, and they display different bone tissue types (Chinnery & Horner, 2007). The humerus mainly contains longitudinal vascular canals, whereas they are more circumferential in the tibia. The latter differs from the vascular pattern of Yinlong and Psittacosaurus in which longitudinal and reticular canals are dominated. Seven LAGs are observed in the tibia, femur, and humerus but no evidence of an avascular external fundamental system, suggesting the sampled individual of Cerasinops is not fully grown.

On the other hand, Protoceratops is a small quadrupedal coronosaur neoceratopsian from the Upper Cretaceous of East Asia, falling outside ceratopsoids (Morschhauser et al., 2018b). Unlike the bone microstructure of Yinlong, Protoceratops displays a prevailing longitudinal vascularization pattern and is mainly comprised of alternating zones of parallel-fibered and woven bone in their humeri, femora, and tibiae (Fostowicz-Frelik & Słowiak, 2018). Bone tissue of Protoceratops fibula is more similar to those of Yinlong. LAGs and annuli were only observed in the fibula of Protoceratops. The level of bone remodeling in the fibula is higher than in the tibia, contrasting to those of basal ornithopod dinosaurs such as Jeholosaurus (Han et al., 2020). A large density of Sharpey’s fibers was observed in the frill and limb bones of Protoceratops (Fostowicz-Frelik & Słowiak, 2018), but they are rare in the bone tissue of Yinlong.

Bone histology of ceratopsids is best known in the chasmosaurine Triceratops, which differs from basal ceratopsians in many aspects (Baag & Lee, 2022; de Rooij et al., 2024). Limb bones consist of a mix of longitudinal and circumferential vascular canals. Alternating zones of parallel-fibered and woven bone are clearly seen in limb elements of Triceratops. Parallel-fibered bone appeared in the juvenile stage of Triceratops. Bone remodeling in Triceratops happened in a much earlier stage than early ceratopsians, whereas LAGs are observed only in a very late stage (de Rooij et al., 2024). In chasmosaurines Utahceratops and Kosmoceratops, vascular canals are longitudinal and reticular, and no LAGs are found in sampled juvenile and subadult individuals (Levitt, 2013). However, LAGs seem to be commonly preserved in centrosaurines (Reizner, 2010; Hedrick et al., 2020). More than five LAGs are present in the tibia of Centrosaurus which is similar in size to the samples of Utahceratops and Kosmoceratops (Levitt, 2013). A maximum of six LAGs are preserved in a rib of Avaceratops and five LAGs are observed near the periphery of the femur in Yehuecauhceratops (Hedrick et al., 2020). As many as 18 LAGs are preserved in the femur of Pachyrhinosaurus (Erickson & Druckenmiller, 2011). The vascular canal pattern of Centrosaurus is different from that of other ceratopsians by the cycled longitudinal and radial orientation in the tibia of Centrosaurus. Histological ontogenetic variation of the centrosaurine ceratopid Einiosaurus has been well studied based on 16 individuals. The tibiae consist of fibrolamellar bone made up of reticular and longitudinal osteons as in basal ceratopsians Yinlong. Radial vascular canals are observed in the early stage (>1 year old). Bone remodeling began after 1 year old, which is earlier than that of Yinlong. In addition, the growth rate of Einiosaurus is highest at 3 years old, which is earlier than those of Yinlong and Psittacosaurus (Erickson et al., 2009). However, the largest size of Einiosaurus is still unknown because the sample individuals were suggested to be less than 6 years old according to the number of LAGs.

LAGs are clear and common in all bone elements of Yinlong, as in Psittacosaurus (Erickson & Tumanova, 2000; Zhao et al., 2013, 2019) and many other dinosaur bones (Chinsamy-Turan, 2005; Han et al., 2020), whereas they are not obvious in more derived ceratopsian Koreaceratops (Baag & Lee, 2022) and Protoceratops and also some ornithopods (e.g., Dysalotosaurus) and many sauropods (Sander et al., 2004; Hübner, 2012). In ceratopsids and sauropods, LAGs only appear in the very late ontogenetic stage (Sander et al., 2011; de Rooij et al., 2024). The regular development of LAGs in large body size of dinosaurs was interpreted as high seasonal stress due to higher food demands, migration, and altricial breeding behavior (Hübner, 2012).

In summary, during the evolution of ceratopsians, there is a discernible augmentation in growth rate, accompanied by an earlier attainment of sexual maturity. Additionally, both bone remodeling and secondary remodeling were accelerated and intensified. There was a decrease in LAG deposition, whereas bone zonation became more pronounced and extensive. Although the body size of ceratopsians continued to increase throughout their evolution, their lifespan did not correspondingly extend, as indicated by the number of preserved LAGs. Therefore, the increase in growth rate likely plays a significant role in the evolution of the substantial body size observed in ceratopsians. The absence of LAGs during the early ontogenetic stages of large-sized ceratopsid dinosaurs suggests rapid and uninterrupted growth. The evolution of bone microstructure in ceratopsians parallels that of sauropods and ornithopods, where LAGs are typically preserved across all growth stages in early-diverging sauropodomorphs and ornithopods, but appear later (in subadult or adult stages) in late-diverging sauropods and hadrosaurs (Klein & Sander, 2008; Cerda et al., 2017; Han et al., 2020; Wosik & Evans, 2022). In contrast, non-avian theropod dinosaurs exhibit LAGs in the early stages of both small and large species, such as Tyrannosaurus, indicating an interrupted growth rate even during juvenile stages (Horner & Padian, 2004; Carr, 2020). D’Emic et al. (2023) argued that the large size of non-avian theropods was the result of both increasing growth rate and duration. These findings provide new information on the growth strategy underlying the evolution of ceratopsians.

Conclusions

This study investigates the bone histology of the earliest ceratopsian Yinlong downsi for the first time, and the results show that the cortex of limb bone mainly consists of fibrolamellar bone that is interrupted by LAGs and parallel-fibered bone. This is similar to Psittacosaurus, whereas in derived ceratopsians, there is usually no clear LAGs and bone remodeling happens earlier. The maximum growth rate of Yinlong is faster than extant reptiles but slower than extant mammals and avians. The growth curve of Yinlong suggests that it reached sexual maturity at about 6 years old, which is earlier than Psittacosaurus, but later than that of late-diverging ceratopsians. It seems that the large size of ceratopsids was a result of fast growth rates and they reached sexual maturity earlier than their ancestors.

Supplemental Information

Supplemental Information 1 Sactional position of Yinlong downsi and Age estimation calculated by using the extrapolation method. .

Supplemental Information 2 Comparison of bone histology in ceratopsian dinosaurs.

We thank the members of the Sino-American expedition team for collecting the fossils studied herein, and Yuzheng Ke and Rui Wu for making thin sections. We thank the editor Blanca Moncunill-Solé and the two anonymous reviewers for their careful review and very useful comments on this manuscript.

Additional Information and Declarations

Competing Interests

Author Contributions

Data Availability

The authors declare that they have no competing interests.

Fenglu Han conceived and designed the experiments, performed the experiments, analyzed the data, prepared figures and/or tables, authored or reviewed drafts of the article, and approved the final draft.

Qi Zhao conceived and designed the experiments, performed the experiments, authored or reviewed drafts of the article, and approved the final draft.

Jinfeng Hu performed the experiments, analyzed the data, prepared figures and/or tables, and approved the final draft.

Xing Xu conceived and designed the experiments, performed the experiments, authored or reviewed drafts of the article, and approved the final draft.

The following information was supplied regarding data availability:

The raw data is available in the Supplemental Files.

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
