# Peer review of "Bone histology and growth curve of the earliest ceratopsian Yinlong downsi from the Upper Jurassic of Junggar Basin, Northwest China"

_PeerJ, doi:10.7717/peerj.18761_

## Round 0.1 · original submission · Major Revisions

Dear authors,

Thank you for your contribution. I have read the paper along with the referees' reviews, and I agree with their assessment that the study and materials are very interesting but require a review before publication. There are several issues that concern me and need to be addressed before the paper can be accepted:

- The consistency of the histological descriptions and the coherence of the discussion.
- Details regarding the growth curve models.
- The lack of some significant histological images.

Please take into account all the comments, suggestions, and tips provided by the referees (in the letter and attached PDFs) to improve the paper and make the necessary changes. If you decide not to follow some recommendations, please provide a thorough justification in the rebuttal letter.

I look forward to reviewing the revised version of your manuscript. Thank you again for your contribution.

PhD Blanca Moncunill-Solé

Reviewer 1 ·

Basic reporting

The article possess the basic features for a PeerJ submit. The text uses technical and scientific writing for a paleohistological study. The cites are well provided, except for a few lacking. The background for the study is good enough. In general, the text is well-structured. The hypothesis is implicit in the text. The authors needs to avoid the usage of the term "can be".

Experimental design

The study extracts histological data from the ceratopsia Yinlong downsi. The research fills the gap regarding the growth dynamics of this stem ceratopsian.The histological descriptions are good enough in general, although these need to get some more consistence. Additionally, the authors create growth curves using histological data and body size estimations. Nevertheless, no information regarding how these were constructed are given. The authors need to add these data to the text, as a brief description, at least.

Validity of the findings

The results are well suported in general. The comparison between different ceratopsians has a robust and reliable character. However, the authors in some parts of the text get lost to the reader respect to nature of the different stages of the studied samples (i.e., it is not in clear the bone matrix composition in the differents stages of the individuals). Finally, it is encourage to the authors to provide some more explanations regarding their findings (i.e., the occurrence of LAGs in early stages and moderate growth rate in comparison with crown groups of ceratopsians).

Annotated reviews are not available for download in order to protect the identity of reviewers who chose to remain anonymous.

Reviewer 2 ·

Basic reporting

See attached

Experimental design

See attached

Validity of the findings

See attached

Additional comments

The authors present a description of Yinlong bone histology based on a large number of specimens and put its histological characteristics in the context of ceratopsian histology as a whole. This is an interesting paper and will be a valuable addition to the literature. I have some larger comments followed by a line-by-line set of comments. I am happy to review a revised version of this manuscript.

Since Yinlong is such an early diverging ceratopsian, it doesn’t make sense to me to just compare it to other ceratopsians in the introduction. The authors go into this in the discussion somewhat where they compare it with ornithopods, but since it is a key intermediate taxon between ceratopsians and their outgroups, it would be perhaps valuable to set up the study that way from the beginning and use Yinlong to elucidate clade-wide trends both within and outside of Ceratopsia similar to what is mentioned in lines 294-98. This would make the study interesting to workers who do not specialize on ceratopsians.

The authors examined ribs, fibulae, and tibiae in a variety of animals giving them the ability to compare histologic patterns across these different bones. I think this would be a great opportunity for an important contribution here, but the authors only touch on the differences in histologic patterns across the bones in the discussion a little bit. It would be great for the authors to compare the differences within individuals more fully and also compare the within individual differences to other species where individuals have had multiple bones sectioned. For example, the authors get into this when they say that ribs are best for counting LAGs in juveniles, but it seems like all of the bones are sort of treated the same for the rest of the manuscript. This should be expanded and would be useful to workers who are interested in paleohistology more generally.

When discussing the different ceratopsian species in the discussion, it would be helpful to get an idea of relative size of animals as well as phylogenetic position. How big was the Koreaceratops that was sectioned? Is it an adult? Etc.

The authors primarily discuss Triceratops and Einioceratops when discussing ceratopsid histology. However, Pachyrhinosaurus, Centrosaurus, and Avaceratops well known as are some other species. I have added some references that might be useful to add in the line-by-line comments.

I commend the authors for showing sections under both normal and cross-polarized light. That makes it very easy to examine the author’s assertions.

Fig 2. Can you change the color of the arrows to be more noticeable?

Fig 10. Where did the regression lines for the non-Yinlong taxa come from? It should say in the figure caption. Also, how were these regressions calculated? Is this just an OLS regression? Add this to the methods.

Line by Line:
Line 27: I don’t think ‘was supposed’ is the right wording here. Perhaps, ‘was suggested’
Line 37: ‘successful, large’ add comma
Line 41: Are all early-diverging ceratopsians really well studied? Some like the ones you mention are pretty well studied. However, Liaoceratops could use restudy as could many other than Psittacosaurus and Auroraceratops
Line 50: I would think of Protoceratops and Koreaceratops as being relatively early diverging. Maybe they belong in the preceding paragraph rather than in this one with more derived ceratopsids? Note that Protoceratops also showed social behavior, which may be useful at fitting it into the above paragraph (https://round-lake.dustinice.workers.dev:443/https/doi.org/10.1666/11-008.1)
Line 51: Also Pachyrhinosaurus (https://round-lake.dustinice.workers.dev:443/https/doi.org/10.1080/08912963.2010.546856), Centrosaurus (Levitt, 2013), Avaceratops, and Yuehecauhceratops (https://round-lake.dustinice.workers.dev:443/https/doi.org/10.1002/ar.24099). I would add them in here as well.
Line 62: ‘stage, but’ add comma
Line 64: perhaps ‘earliest known ceratopsian’
Line 68: ‘Neoceratopsia’
Line 77: What do you mean ‘as many as 10’?
Line 81-83: Grammar
Line 77 and 84: Use either 10 or ‘ten’ depending on journal requirements. However, be consistent throughout.
Line 86: ‘observations’
Line 92-93: Is this really a safe assumption? Probably the histologic patterns across these early diverging ceratopsians will not be different enough to make much of a difference, but what is the advantage of including these specimens if they have no skull? Do they fill in a size gap that is important? I think they should be given a different shape on the figures so that they can be distinguished as perhaps not being Yinlong.
Line 95: Most was at IVPP? Not worth mentioning unless you say where the others were prepped.
Line 112–114: Break this sentence up. End the first part after Fig. 1 and then, ‘We preferred…’. You might note examining ribs was done by Waskow and Sander (2014) in sauropods due to long bone remodeling (expanded discussion on that topic in that paper) and Fostowicz-Frelik and Slowiak (2018), Hedrick et al. (2020), de Rooij et al. (2024) in ceratopsians. That would get you away from ‘preferred’ which is not scientific sounding.
Line 124: You only have one equation here so ‘equation’
Line 142: No need to capitalize these words
Line 144: You explained in the methods that you preferred to count LAGs in ribs for small individuals, but here say that descriptions of each stage (including the early juvenile stage?) are based on tibiae.
Line 148–149: Could be identified as the early juvenile stage or do you use the histologic patterns here to define the early juvenile stage?
Line 150: ‘and there are only a few…’
Line 151: What do you mean by ‘weak’ LAGs?
Line 157: ‘clearly visible’ rather than ‘clearly shown’
Line 161: Or there is variation in growth across bones where the LAG was not deposited in the tibia, but was in the fibula.
Line 165: ‘The late juvenile stage…’
Line 169: ‘primary’ shouldn’t be capitalized.
Line 171: ‘visible’ rather than ‘shown’
Line 175: ‘in most of the cortex..’
Line 181: ‘suggesting the individual was a minimum of five years old…’
Line 184: ‘were sampled’ Be careful of tense changes in the results here. Check you are either in the present or past tense, but don’t mix and match.
Line 194: Remove bolding
Line 198: Given the presence of double LAGs, perhaps the age estimates are an overestimate and two LAGs were deposited in a single year? You touch on this in the discussion and that could be expanded a bit.
Line 205: Capitalize ‘Haversian’
Line 211: Remove bolding
Line 236: ‘scarce’
Line 238–39: Does this suggest a EFS? If not, why not?
Line 252: ‘all of the…’
Line 251–263: There is no real comparison here so this strikes me as a summary of the results rather than discussion. Add in comparisons here with other taxa.
Line 259–263: Is this true across all ontogenetic stages? How does this compare with other taxa? This would be important to a broader audience.
Line 267–268: I think you can expand on the discussion of double LAGs and what it means for counting LAGs to get minimum ages.
Line 271–272: Hedrick et al. (2016) discusses this exact thing in a Psittacosaurus https://round-lake.dustinice.workers.dev:443/https/doi.org/10.1002/ar.23363
Line 297: ‘iguanodontians’
Line 297–298: Could this be related to body size? Early diverging taxa are smaller.
Line 304: ‘alligators’
Line 307-08: How are you distinguishing closely spaced LAGs here from an EFS?
Line 326: add period
Line 325: Space between P.l
Line 328: ‘vascularization that is suggested to have…’
Line 328-31: This paragraph needs some work. I’m not sure what the point is and it is hard to tell which species have what.
Line 365–379: This is a good start, but doesn’t discuss Pachyrhinosaurus, Centrosaurus, or Avaceratops. That would allow expansion of how ceratopsid bone histology compares with a basal ceratopsian like Yinlong
Line 381–393: These paragraphs could be worked into one of the above paragraphs. As written, they seem like afterthoughts.
Line 391–393: This could also just be because large dinosaurs have to get large quickly so they don’t deposit LAGs until their growth slows down. It could be an interesting discussion point to increase the relevance of the paper to all paleohistological workers.
All Fig captions. ‘arrows’ rather than ‘allows’
All Fig captions. Just ‘light’ rather than ‘lights’
Table 1: What do the parenthetical numbers mean by ‘visible LAGs’?

Annotated reviews are not available for download in order to protect the identity of reviewers who chose to remain anonymous.

---

## Round 0.2 · Minor Revisions

Dear Authors,

Thank you for submitting the revised version of your paper. I agree with the referees that the manuscript has improved significantly. While the content is correct, we have noticed several typographical and typesetting errors that should be addressed before the paper can be accepted for final publication.

These issues are highlighted in the referees' comments, and I have also attached a PDF with the suggested corrections marked. These changes will enhance the readability of the paper. Additionally, I have suggested some adjustments to the figures. Currently, some figures lack scales, and the placement of the letters indicating figure parts is inconsistent. In bone histology papers, the quality and accuracy of the images are crucial, so extra care must be taken to ensure they meet the required standards.

I hope you can address these points promptly.

Best regards,
Dr. Blanca Moncunill-Solé

Reviewer 1 ·

Basic reporting

The authors cover all the issues marked in the first round of reviews and response all the comments with justified answers. In this sense, the article is ready to be released.

Experimental design

The authors add and modified the information required to fulfill with this field.

Validity of the findings

The authors add and modified the information required to fulfill with this field.

Reviewer 2 ·

Basic reporting

See below

Experimental design

See below

Validity of the findings

See below

Additional comments

The authors have done an excellent job at responding to a previous round of reviews and I think that this is an important study in terms of presenting the histology of Yinlong, but also in synthesizing the state of what is known about ceratopsian growth and histology.

There are a few minor changes that I think should be incorporated prior to publication, which amount to word changes for grammar/clarity:

Line 28: When you say ceratopsians, do you mean basal ceratopsians?
Line 106: Only two have postcranial elements? Do you mean that only two have only postcranial elements and no skulls associated? They all must have some postcranial material to do histo on.
Line 373: ‘during their evolution’ rather than ‘the’
Line 414: ‘largest known’ rather than ‘known largest’
Line 477: ‘LAGs are preserved’ rather than ‘LAGs is preserved’
Sharpey’s should be capitalized in the figure captions

---

## Round 0.3 · accepted · Accept

Dear Authors,

Congratulations! This final version has addressed the latest referees' comments succesfully. In my view, the review process has significantly enhanced the quality of your paper and further solidified the results. Each section is clearly and thoroughly explained, and the findings are outstanding! Thank you for your time, dedication, and patience throughout the review process.

I wish you all the best in your future research endeavors and career.

Best regards,
PhD Blanca Moncunill-Solé